# Message Passing and Metabolism

**DOI:** 10.3390/e23050606

**Published:** 2021-05-14

**Authors:** Thomas Parr

**Affiliations:** Wellcome Centre for Human Neuroimaging, Queen Square Institute of Neurology, University College London, London WC1N 3AR, UK; thomas.parr.12@ucl.ac.uk

**Keywords:** message passing, metabolism, Bayesian, stochastic, non-equilibrium, master equations

## Abstract

Active inference is an increasingly prominent paradigm in theoretical biology. It frames the dynamics of living systems as if they were solving an inference problem. This rests upon their flow towards some (non-equilibrium) steady state—or equivalently, their maximisation of the Bayesian model evidence for an implicit probabilistic model. For many models, these self-evidencing dynamics manifest as messages passed among elements of a system. Such messages resemble synaptic communication at a neuronal network level but could also apply to other network structures. This paper attempts to apply the same formulation to biochemical networks. The chemical computation that occurs in regulation of metabolism relies upon sparse interactions between coupled reactions, where enzymes induce conditional dependencies between reactants. We will see that these reactions may be viewed as the movement of probability mass between alternative categorical states. When framed in this way, the master equations describing such systems can be reformulated in terms of their steady-state distribution. This distribution plays the role of a generative model, affording an inferential interpretation of the underlying biochemistry. Finally, we see that—in analogy with computational neurology and psychiatry—metabolic disorders may be characterized as false inference under aberrant prior beliefs.

## 1. Introduction

Common to many stochastic systems in biology is a sparse network structure [1,2,3,4]. In the nervous system, this manifests as many neurons that each synapse with a small subset of the total number [5]. In biochemistry, similar network structures exist, in which each chemical species reacts with a small number of other chemicals—facilitated by specific enzymes [6]. In both settings, the ensuing dynamics have parallels with techniques applied in the setting of probabilistic inference, where the sparsity is built into a (generative) statistical model that expresses how observable data are caused by latent or hidden states. Inversion of the model, to establish the most plausible causes for our observations, appeals to the conditional dependencies (and independencies) between its constituent hidden variables [7,8]. This has the appearance of message passing between nodes in a network of variables, with messages passed between nodes representing variables that conditionally depend upon one another [9].

The homology between inferential message passing and sparse dynamical systems is central to active inference—a theoretical framework applied primarily in the neurosciences [10]. Active inference applies to stochastic dynamical systems whose behaviour can be framed as optimisation of an implicit model that explains the inputs to that system. More specifically, it treats the dynamics of a system as a gradient flow on a marginal likelihood (a.k.a., model evidence), that is the minimum of a free energy functional [11]. The internal dynamics of a system are then seen as minimising free energy to find the marginal likelihood, which itself is maximized through acting to change external processes so that the system inputs become more probable [12].

In the brain sciences, active inference offers a principled approach that underwrites theoretical accounts of neuronal networks as performing Bayesian inference [13,14]. However, the same mathematics is also applicable to other biotic systems—as has been shown in the context of self-organisation and morphogenesis [15]—and even to non-biological systems [16]. This paper attempts to find out how far we can take this approach in the biochemical domain. This means an account of metabolic principles in terms of generative models, their constituent conditional dependencies, and the resulting probabilistic dynamics.

Part of the motivation for focusing on metabolism is that it calls for a slightly different formulation of stochastic dynamics to the Fokker–Planck formalisms [17] usually encountered in active inference [18]. As chemical species are categorical entities (e.g., glucose or fructose), discrete probability distributions, as opposed to continuous densities, are most appropriate in expressing our beliefs about chemical reactions. Systems of chemical reactions then offer useful concrete examples of a slightly different form of stochastic dynamics to those explored using Fokker–Planck equations. However, despite attempting to establish a construct validity in relation to metabolism, the primary focus of this paper is not biochemical. It is on the applicability of probabilistic dynamics, of the sort employed in active inference, to systems that are not made up of neuronal networks. Specifically, it is on the emergence of networks in stochastic dynamical systems, under a particular generative model, and upon the interpretation of the network dynamics as inferential message passing.

The argument of this paper can be overviewed as follows. Given an interpretation of a steady state as a generative model, the behaviour of stochastic systems that tend towards that distribution can be interpreted as inference. When such systems are formulated in terms of master equations, they have the appearance of gradient flows on a free energy functional—the same functional used to score approximate Bayesian inference schemes. In practice, we may be interested in high-dimensional systems, for which the distribution can be factorized into a set of lower-dimensional systems. In this setting, a mean-field approximation can be applied such that we only need work with these lower-dimensional marginal distributions. When the implicit generative model is sufficiently sparse, steady state can be achieved through dynamics that involve sparse coupling between the marginal distributions—analogous to inferential message passing schemes in machine learning. Under certain assumptions, these probabilistic dynamics have the appearance of chemical kinetics, licensing an interpretation of some chemical reactions—including biological, enzymatic, reactions—as if the chemical species were engaged in a form of (active) inference about one another. An implication of this interpretation is that metabolic pathologies can be framed in terms of the implicit generative model (i.e., steady state) they appear to be solving.

The main sections of this paper are organized as follows. Section 2 outlines probabilistic dynamics of a categorical sort, and the relationship between these dynamics and the notion of a generative model. This includes the use of mean-field assumptions and the construction of dynamical systems with a particular steady state in mind. Section 3 applies these ideas in the setting of biochemistry, relating the probability of being in each configuration to the concentrations of the associated chemicals. Here, under certain assumptions, we see the emergence of the law of mass action, and the Michaelis–Menten equation—standard results from biochemistry. Section 4 offers an example of a biochemical network, based on the kinetics developed in the previous sections. This paper then concludes with a discussion of the relationship between message passing of the sort found in biochemical and neurobiological systems.

## 2. Probabilistic Dynamics

### 2.1. Free Energy and Generative Models

Bayesian inference depends upon an important quantity known as a marginal likelihood. This tells us how probable the data we have observed are, given our model for how those data are generated. As such, it can be thought of as the evidence afforded to a model by those data [19]. For stochastic dynamical systems that have a (possibly non-equilibrium) steady state, the partition function at this steady state can be interpreted as if it were a marginal likelihood [12]. This lets us think of such systems as ‘self-evidencing’ [20], in the sense that they tend over time towards a high probability of occupancy for regions of space with a high marginal likelihood, and low probability of occupancy for regions with a low marginal likelihood.

The term ‘marginal’ refers to the summation, or integration, of a joint probability distribution with respect to some of its arguments, such that only a subset of these arguments remains in the resulting marginal distribution. In Bayesian statistics, we are normally interested in finding the marginal likelihood, under some model, of data (*y*). This is a common measure of the evidence the data affords to the model. However, the model includes those variables (*x*) that conspire to generate those data. These are variously referred to as hidden or latent states. It is these states that must be marginalized out from a joint distribution.

Variational inference reframes this summation (or integration) problem as an optimisation problem [8,21], in which we must optimize an objective functional (function of a function) whose minimum corresponds to the marginal likelihood. This functional is the variational free energy, defined as follows:(1)$$\begin{array}{cc} F\left[y,P(x,\tau |y)\right] & ≜E_{P(x,\tau |y)}\left[\ln P(x,\tau |y)-\ln P(x,y,\infty )\right]\\ \ln P(y,\infty ) & =-\underset{P(x,\tau |y)}{\min}F\left[y,P(x,\tau |y)\right]\\ P\left(x,\infty |y\right) & =\underset{P(x,\tau |y)}{\arg\min}F\left[y,P(x,\tau |y)\right] \end{array}.$$

The notation *P*(*x*,*τ*) should be read as the probability that a random variable *X* takes a value *x* at time *τ*, consistent with conventions in stochastic thermodynamics [22]. The symbol E means the expectation, or average, under the subscripted probability distribution. Equation (1) says that, if free energy is minimized with respect to a time-dependent probability density, then self-evidencing can proceed through minimisation of free energy with respect to *y*. In neurobiology, minimisation of an upper bound on negative log model evidence is associated with synaptic communication [10]. This neural activity ultimately results in muscle contraction, which causes changes in *y* [23,24].

An important perspective that arises from Equation (1), and from an appeal to free energy minimisation, is the association between a steady-state distribution and a generative model. The generative model can be written in terms of prior and likelihood distributions whose associated posterior and model evidence (marginal likelihood) are the minimizer of and minimum of the free energy, respectively:(2)$$\begin{array}{ll} P\left(x,y,\infty \right) & =\underset{\mathrm{Likelihood}}{\underbrace{P\left(y,\infty |x\right)}}\underset{\mathrm{Prior}}{\underbrace{P\left(x,\infty \right)}}\\ & =\underset{\mathrm{Posterior}}{\underbrace{P\left(x,\infty |y\right)}}\underset{\mathrm{Model}\;\mathrm{evidence}}{\underbrace{P\left(y,\infty \right)}} \end{array}.$$

The key insight from Equation (2) is that, when free energy changes such that it comes to equal the negative log model evidence, the requisite evolution of the time-dependent conditional distribution tends towards a posterior probability. The implication is that when we interpret the partition function (i.e., marginal) of the steady-state distribution as if it were model evidence, the process by which the system tends towards its steady state over time has an interpretation as Bayesian inference.

### 2.2. Master Equations

This paper’s focus is upon probabilistic dynamics of a categorical sort. This deviates from recent accounts [12,18,25] of Bayesian mechanics in terms of Fokker–Planck equations, which describe the temporal evolution of probability density functions. However, master equations [26,27] afford an analogous expression for the dynamics of categorical probability distributions. The form of these equations can be motivated via a Taylor series expansion:(3)$$\begin{array}{cc} P(x,\tau ) & =E_{P(z,t)}\left[P(x,\tau |z,t)\right]\\ P\left(x,\tau |z,t\right) & =\underset{\delta_{xz}}{\underbrace{P(x,t|z,t)}}+\partial_{t}P(x,t|z,t)\Delta \tau +O(\Delta \tau^{2})\\ \Delta \tau & =\tau -t \end{array}.$$

Equation (3) uses the Kronecker delta function. This expresses the fact that, given *X* = *z* at time *t*, the probability that it is equal to *x* at time *t* is one when *x* is equal to *z*, and is zero otherwise. Substituting the Taylor series expansion of the second line into the first line gives us:(4)$$\begin{array}{l} P(x,\tau )=P(x,t)+E_{P(z,t)}\left[\partial_{t}P(x,t|z,t)\right]\Delta \tau +O(\Delta \tau^{2})\\ \;\Rightarrow \\ \partial_{\tau }P(x=i,\tau )=\sum_{j}L_{ij}P(z=j,\tau )\\ \;L_{ij}≜\partial_{\tau }P(x=i,\tau |z=j,\tau ) \end{array}.$$

The transition rate matrix **L** determines the rate at which probability mass moves from one compartment to another. The dynamics of a time-dependent distribution are as follows. Assuming that it is a categorical distribution, whose sufficient statistics comprise a vector of probabilities **p**:(5)$$\begin{array}{cc} P\left(x,\tau \right) & =Cat\left(p(\tau )\right)\\ \partial_{\tau }p(\tau ) & =Lp(\tau ) \end{array}.$$

The elements of **p** are the probabilities of the alternative states *x* may assume and must sum to one. Now that we have an expression for a time-dependent probability distribution, how do we link this back to the steady state of the free energy minimum from Equation (1)? One answer to this question comes from recent work that formulates the dynamics of a master equation in terms of a potential function that plays the role of a steady state [28]. This involves the following decomposition of the transition rate matrix:(6)$$\begin{array}{cc} \partial_{\tau }p(\tau ) & =\left(Q-\Gamma \right)\Lambda p(\tau )\\ \Gamma & ≜-\frac{1}{2}\left(A+A^{T}\right)\\ Q & ≜\frac{1}{2}\left(A-A^{T}\right)\\ A & ≜L\Lambda^{-1}\\ \Lambda & ≜diag{\left(p(\infty )\right)}^{-1} \end{array}.$$

The steady-state distribution **p**(∞) is given by the (normalized) right singular vector of **L** whose singular value is zero. This follows directly from a singular value decomposition of Equation (5). Equation (6) decomposes the transition rate matrix into two parts, the first with a skew-symmetric matrix **Q** and the second with a symmetric matrix **Γ**. This construction resembles the Helmholtz decomposition sometimes applied to continuous dynamical systems [29]—where the flow is decomposed into a solenoidal (conservative) and a dissipative component.

We can relate Equation (6) directly to the minimisation of free energy in Equation (1) when **p** is a distribution conditioned upon some input variable *y*. This rests upon a local linear approximation to the gradient of the free energy. Using the notation of Equation (6), free energy and its gradient are:(7)$$\begin{array}{cc} F(p(\tau )) & =p(\tau )\cdot \left(\ln p(\tau )-\ln p(\infty )\right)+z(\infty )\\ z(\infty ) & ≜\ln P(y,\infty )\\ \nabla_{p(\tau )}F(p(\tau )) & =\ln p(\tau )-\ln p(\infty )+1\\ & =1+\ln\Lambda p(\tau )\\ & \approx \Lambda p(\tau )\\ & \Rightarrow \\ \partial_{\tau }p(\tau ) & \approx \left(Q-\Gamma \right)\nabla_{p(\tau )}F(p(\tau )) \end{array}.$$

The approximate equality in the fifth line follows from a Taylor series expansion (around Λ**p**(*τ*) = **1**) of the logarithm in the previous line, truncated after the linear term. This tells us that Equation (6) is, at least locally, a gradient descent on free energy augmented with a solenoidal flow orthogonal to the free energy gradients. Figure 1 illustrates this using a randomly generated transition rate matrix for a system with three possible states. The steady state was identified from the right singular vectors of this matrix, facilitating computation of the free energy. Starting from different locations in the free energy landscape, we see that the combination of the solenoidal and dissipative flows is consistent with a gradient descent on the free energy landscape. The dissipative flow has a similar appearance, while the solenoidal trajectories travel along the free energy gradients, eventually leaving the simplex (violating the conservation of probability).

### 2.3. Mean-Field Models

The above establishes that probabilistic dynamics under a master equation can be formulated in terms of a gradient descent on variational free energy. However, these dynamics appear limited by their linearity. As nonlinear dynamical systems are ubiquitous in biology, it is reasonable to ask what relevance Equation (5) has for living systems. The answer is that, when *x* can be factorized, Equation (5) deals with the evolution of a joint probability distribution. Linear dynamics here do not preclude non-linear dynamics of the associated marginal distributions that deal with each factor of *x* individually. This section unpacks how non-linear behaviour emerges from Equation (5) when we adopt a mean-field assumption. A mean-field assumption [30,31,32] factorizes a probability distribution into a set of marginal probabilities (*Q*):(8)$$\begin{array}{cc} Q(x_{1},\ldots ,x_{N},\tau ) & =\prod_{i}Q(x_{i},\tau )\\ & \Rightarrow \\ \partial_{\tau }Q(x_{i},\tau ) & \approx E_{\prod_{k}Q(x_{k},\tau )}\left[\partial_{\tau }P(x_{i},\tau |x_{1},\ldots ,x_{N},\tau )\right] \end{array}.$$

Equation (8) depends upon the same steps as Equations (3) and (4). It effectively decomposes the dynamics of the joint probability into those of a set of linked compartmental systems. Rewriting this in the form of Equation (5) gives:(9)$$\begin{array}{c} \partial_{\tau }q_{j}^{i}(\tau )=\sum_{klmn\ldots }L_{jklmn\ldots }^{i}q_{k}^{1}(\tau ),\ldots ,q_{s}^{N}(\tau )\\ Q(x_{i},\tau )=Cat\left(q^{i}(\tau )\right) \end{array}.$$

This formulation allows for dynamical interactions between the marginal distributions. Equation (9) can be re-written, using a Kronecker tensor product, to illustrate the savings associated with a mean-field approximation. Here, we see that, although we retain the same number of columns as in our original transition rate matrix, the number of rows reduces to the sum of the lengths of the marginal vectors.
(10)$$\begin{array}{c} q(\tau )=q^{1}(\tau )⊗q^{2}(\tau )⊗q^{3}(\tau )⊗\cdots \\ L^{i}=\left[\begin{array}{cc} \begin{array}{ccc} L_{111\ldots }^{i} & L_{121\ldots }^{i} & \cdots \\ L_{211\ldots }^{i} & L_{221\ldots }^{i} & \\ \vdots & & \ddots \end{array} & \begin{array}{ccc} L_{112\ldots }^{i} & L_{122\ldots }^{i} & \cdots \\ L_{212\ldots }^{i} & L_{222\ldots }^{i} & \\ \vdots & & \ddots \end{array} \end{array}\right]\\ \partial_{\tau }q^{i}(\tau )=L^{i}q(\tau ) \end{array}.$$

This formulation is useful, as we can use it to engineer an **L** that would lead to a desired steady state. We can do this by defining **L** in terms of the components of its singular value decomposition. This involves setting one of the right singular vectors equal to the desired steady state, and setting the associated singular value to zero:(11)$$\begin{array}{c} L^{i}=AS^{i}V^{i}\\ S^{i}=\left[\begin{array}{ccc} 0 & 0 & \cdots \\ 0 & \lambda^{i} & \\ \vdots & & \ddots \end{array}\right],V^{i}=\left[\begin{array}{l} p{\left(\infty \right)}^{T}\\ v^{i}\\ \vdots \end{array}\right]\\ A=\left[\begin{array}{ccc} 1 & a & \cdots \end{array}\right] \end{array}.$$

In what follows, we will assume we are dealing with binary probabilities, such that **q***^i^*(*τ*) is a two-dimensional vector for all *i*. To simplify things, we only concern ourselves with the second column of **A** and second row of **V**:(12)$$L^{i}=\lambda^{i}av^{i}.$$

It is worth noting that this choice can result in there being more than one fixed point. However, some of these will be inconsistent with the simplex that defines allowable probability distributions and can be safely ignored. Others impose limits on the initial conditions of a dynamical system for which Equation (12) is valid. The terms from Equation (12) can be parameterized as follows:(13)$$\begin{array}{cc} a & ={\left[\begin{array}{cc} 1 & -1 \end{array}\right]}^{T}\\ v^{i}p(\infty ) & =0\\ v^{i} & =\beta^{i}\left[\begin{array}{c} \left[\begin{array}{cc} 1 & 1 \end{array}\right]⊗\left[\begin{array}{cc} 1 & 1 \end{array}\right]⊗\left[\begin{array}{cc} 1 & 1 \end{array}\right]\cdots \\ \left[\begin{array}{cc} 1 & 0 \end{array}\right]⊗\left[\begin{array}{cc} 1 & 1 \end{array}\right]⊗\left[\begin{array}{cc} 1 & 1 \end{array}\right]\cdots \\ \left[\begin{array}{cc} 1 & 1 \end{array}\right]⊗\left[\begin{array}{cc} 1 & 0 \end{array}\right]⊗\left[\begin{array}{cc} 1 & 1 \end{array}\right]\cdots \\ \left[\begin{array}{cc} 1 & 0 \end{array}\right]⊗\left[\begin{array}{cc} 1 & 0 \end{array}\right]⊗\left[\begin{array}{cc} 1 & 1 \end{array}\right]\cdots \\ \vdots \\ \left[\begin{array}{cc} 1 & 0 \end{array}\right]⊗\left[\begin{array}{cc} 1 & 0 \end{array}\right]⊗\left[\begin{array}{cc} 1 & 0 \end{array}\right]\cdots \end{array}\right] \end{array}.$$

Here, the **a** vector is assumed to be the same for all factors of the probability distribution. It ensures any change in probability for one of the binary states is combined with an equal and opposite change in probability for the other state. In other words, it ensures the columns of the transition rate matrix sum to zero, consistent with conservation of probability. The **v** (row) vector is parameterized in terms of a weighted sum of row vectors, each of returns a different marginal when multiplied with **p** or **q**. This leads to the following expression for the transition probabilities:(14)$$L^{i}q(\tau )=\lambda^{i}a\left(\beta_{1}^{i}+\beta_{2}^{i}q_{1}^{1}(\tau )+\beta_{3}^{i}q_{1}^{2}(\tau )+\beta_{4}^{i}q_{1}^{1}(\tau )q_{1}^{2}(\tau )+\ldots \right).$$

There may be multiple combinations of the *β* coefficients that satisfy the condition that **p**(∞) is orthogonal to **v***^T^*. The next subsection offers one way in which we can constrain these, based upon the notion of a Markov blanket [33].

### 2.4. Graphical Models and Message Passing

It need not be the case that all variables in a generative model depend upon all others. Before moving to a chemical interpretation of the probabilistic dynamics outlined above, it is worth touching upon their interpretation as message passing when the underlying model is sufficiently sparse. For this, we need the concept of a Markov blanket [33]. A Markov blanket for *x_i_* is the set of variables that render all other variables conditionally independent of *x_i_*. For example, if the Markov blanket of *x_i_* is *x_j_* at steady state, the following relationship holds:(15)$$\begin{array}{c} \left(x_{i}⊥\{x_{k}:k\neq i,k\neq j\}\right)|x_{j}\;\;\;\;\;\;\;\;\;\\ \;\;\;\;\;\;\;\;\;\Rightarrow P\left(x_{i},\infty |\{x_{k}:k\neq i\}\right)=P\left(x_{i},\infty |x_{j}\right) \end{array}.$$

The implication is that any marginals corresponding to variables outside of the Markov blanket, under the steady-state distribution, add no additional information as to the steady state for the variable inside the blanket. As such, we can set the *β* coefficients from Equation (14) to zero for all terms except those exclusively comprising variables in the Markov blanket. Figure 2 shows an example of a generative model, expressed as a normal factor graph [34], that illustrates an interpretation of the associated dynamics as message passing among the posterior marginals.

To summarize, so far, the dynamics of a categorical probability distribution can be formulated in terms of a master equation. The steady state of these dynamics is interpretable as the minimizer of a free energy functional of the sort used in variational Bayesian inference. Locally, the dynamics of a master equation can be formulated as a gradient descent on the same variational free energy. Crucially, free energy is a functional of a generative model. Starting from this model, we can construct master equations that lead to steady states consistent with that model. One way of doing this is to specify the components of a singular value decomposition of a probability transition matrix such that the singular value, corresponding to the right singular vector parallel to the steady state, is zero. For systems with many components, it is often more efficient to deal with a mean-field approximation of the dynamics. This lets us formulate the dynamics of marginal distributions for each variable without reference to the probability of variables outside of the Markov blanket of those variables. The next section highlights the points of contact between these mean-field dynamics and established concepts in biochemistry. These include the law of mass action, reaction networks, and Michaelis–Menten enzyme kinetics.

## 3. Biochemical Networks

### 3.1. The Law of Mass Action

This section starts by relating the mean-field dynamics of the previous section to the law of mass action, which specifies the relationship between the rate of a chemical reaction and the concentrations of the chemical species involved in that reaction [35,36,37]. A reversible chemical reaction is expressed as follows:(16)$$\sum_{i}\sigma_{i}S_{i}⇌\sum_{i}\rho_{i}S_{i}.$$

Here, *S* stands in for the different chemical species (indexed by the subscript), and *σ* and *ρ* for the stoichiometric coefficients (i.e., the number of molecules of *S* used up by, or produced by, a single reaction taking place, respectively). The symbol between substrates and products indicates a reversible transition between the substrates and products.

Our challenge is to express Equation (16) in terms of the joint distribution of chemical species at steady state (i.e., a generative model), and then to find an appropriate master equation to describe the route to this steady state. The following shows the form of a steady state for a system with two substrates and two products:(17)$$\begin{array}{cc} P\left(S_{1}⊗S_{2},\infty \right) & =Cat\left(\left[\begin{array}{l} \alpha \\ 0\\ 0\\ 1-\alpha \end{array}\right]\right)\\ P\left(S_{3}⊗S_{4},\infty |S_{2}\right) & =Cat\left(\left[\begin{array}{cc} \begin{array}{c} 0\\ 0\\ 0\\ 1 \end{array} & \begin{array}{c} 1\\ 0\\ 0\\ 0 \end{array} \end{array}\right]\right) \end{array}.$$

Intuitively, the first probability distribution tells us that the only plausible configurations of the two substrate molecules are both present and both absent. The second distribution says that, if the second (and therefore the first) substrates are present, the products are both absent. However, if the substrates are absent, the products are present. As all variables of Equation (17) are conditionally dependent upon all other variables, the resulting master equations must depend upon the marginals for all variables. In selecting **v**, we can group these depending upon which side of the reaction they occupy:(18)$$\begin{array}{cc} v^{i} & =\beta \left[\begin{array}{cc} 1 & 1 \end{array}\right]⊗\left[\begin{array}{cc} 1 & 1 \end{array}\right]⊗\left[\begin{array}{cc} 1 & 0 \end{array}\right]⊗\left[\begin{array}{cc} 1 & 0 \end{array}\right]\\ & -\left[\begin{array}{cc} 1 & 0 \end{array}\right]⊗\left[\begin{array}{cc} 1 & 0 \end{array}\right]⊗\left[\begin{array}{cc} 1 & 1 \end{array}\right]⊗\left[\begin{array}{cc} 1 & 1 \end{array}\right]\\ \beta & ≜\left(\frac{\alpha }{1-\alpha }\right)\\ & \Rightarrow \\ \left[\begin{array}{l} \partial_{\tau }q_{1}^{1}(\tau )\\ \partial_{\tau }q_{1}^{2}(\tau )\\ \partial_{\tau }q_{1}^{3}(\tau )\\ \partial_{\tau }q_{1}^{4}(\tau ) \end{array}\right] & =\lambda \left[\begin{array}{l} \beta q_{1}^{3}(\tau )q_{1}^{4}(\tau )-q_{1}^{1}(\tau )q_{1}^{2}(\tau )\\ \beta q_{1}^{3}(\tau )q_{1}^{4}(\tau )-q_{1}^{1}(\tau )q_{1}^{2}(\tau )\\ q_{1}^{1}(\tau )q_{1}^{2}(\tau )-\beta q_{1}^{3}(\tau )q_{1}^{4}(\tau )\\ q_{1}^{1}(\tau )q_{1}^{2}(\tau )-\beta q_{1}^{3}(\tau )q_{1}^{4}(\tau ) \end{array}\right] \end{array}.$$

The final line follows from the previous lines via Equation (12). The probabilities that the chemical species is absent (subscript 2) have been omitted as they are simply the complement of the probabilities that they are present (subscript 1). Note that the marginals (at steady state) that result from these dynamics are not consistent with the marginals of the generative model. This is due to the violation of the mean-field assumption. We can correct for the discrepancy by raising the numerator of *β* to a power of the number of substrates, and the denominator to the power of the number of reactants. This correction is obtained by setting Equation (14) to zero when the marginals are consistent with those at steady state and solving for the *β* coefficients. In addition, these kinetics conserve the summed probability of species being present (i.e., they conserve mass) so cannot achieve the steady state from Equation (17) unless the initial conditions include a summed probability of presence of 1.

By interpreting the probabilities as the proportion of the maximum number (*N*) of molecules of each species, and dividing these by the volume (*V*) in which they are distributed, Equation (18) can be rewritten in terms of chemical concentrations (**u**):(19)$$\begin{array}{cc} \left[\begin{array}{l} \partial_{\tau }u_{1}(\tau )\\ \partial_{\tau }u_{2}(\tau )\\ \partial_{\tau }u_{3}(\tau )\\ \partial_{\tau }u_{4}(\tau ) \end{array}\right] & =\left[\begin{array}{l} \kappa_{2}u_{3}(\tau )u_{4}(\tau )-\kappa_{1}u_{1}(\tau )u_{2}(\tau )\\ \kappa_{2}u_{3}(\tau )u_{4}(\tau )-\kappa_{1}u_{1}(\tau )u_{2}(\tau )\\ \kappa_{1}u_{1}(\tau )u_{2}(\tau )-\kappa_{2}u_{3}(\tau )u_{4}(\tau )\\ \kappa_{1}u_{1}(\tau )u_{2}(\tau )-\kappa_{2}u_{3}(\tau )u_{4}(\tau ) \end{array}\right]\\ u_{i}(\tau ) & ≜V^{-1}Nq_{1}^{i}(\tau )\\ \kappa_{1} & ≜\lambda N^{-1}V\\ \kappa_{2} & ≜\lambda \beta N^{-1}V\\ \beta & =\frac{\alpha^{2}}{{\left(1-\alpha \right)}^{2}} \end{array}.$$

Equation (19) uses the ‘corrected’ *β* coefficient. A simulation of this reaction is shown in Figure 3, represented both in terms of the evolution of probability and as chemical concentrations. Note the gradual transition from substrates to products, under a generative model in which the latter are more probable. The free energy of this reaction can be seen to monotonically decrease, highlighting the consistency with the dynamics of Equation (7) despite the mean-field assumptions.

In chemical systems, the rate of change of some reactants can depend non-linearly on the concentration of those reactants. We can take a step towards this relationship as follows. If we then stipulate that two (or more) of the chemical species are the same, we can re-express this, with suitable modification of the constants, as:(20)$$\begin{array}{cc} 2S_{1} & ⇌S_{3}+S_{4}\\ \begin{array}{c} u_{1}(\tau )\propto q_{1}^{1}(\tau )+q_{1}^{2}(\tau )\\ q_{1}^{1}(0)=q_{1}^{2}(0) \end{array}\} & \Rightarrow \\ \left[\begin{array}{l} \partial_{\tau }u_{1}(\tau )\\ \partial_{\tau }u_{3}(\tau )\\ \partial_{\tau }u_{4}(\tau ) \end{array}\right] & =\left[\begin{array}{l} 2\kappa_{2}u_{3}(\tau )u_{4}(\tau )-2\kappa_{1}u_{1}{\left(\tau \right)}^{2}\\ \kappa_{1}u_{1}{\left(\tau \right)}^{2}-\kappa_{2}u_{3}(\tau )u_{4}(\tau )\\ \kappa_{1}u_{1}{\left(\tau \right)}^{2}-\kappa_{2}u_{3}(\tau )u_{4}(\tau ) \end{array}\right] \end{array}.$$

This brings an autocatalytic element to the reaction, allowing the substrate to catalyse its own conversion to the reaction products. Generalising the above, we can express the law of mass action [35,36,37] for the generic reaction in Equation (16) as:(21)$$\partial_{\tau }u_{i}(\tau )=(\rho_{i}-\sigma_{i})\left(\kappa_{1}\prod_{j}u_{j}{\left(\tau \right)}^{\sigma_{j}}-\kappa_{2}\prod_{j}u_{j}{\left(\tau \right)}^{\rho_{j}}\right).$$

This subsection started with from the mean-field master equation developed in Section 2 and illustrated how, under certain conditions, the law of mass action for chemical systems can be obtained. This application of a master equation to chemical dynamics should not be confused with the chemical master equation, detailed in Appendix A [38]. The key steps were (i) the specification of a generative model for which the marginal probabilities that each chemical species is present sum to one and (ii) the choice of right singular vector, orthogonal to the resulting joint distribution, for the transition rate matrix.

### 3.2. Reaction Networks

In the previous two subsections, all chemical species were assumed to participate in a single reaction. Here, we extend this, such that we can see how the message passing from Figure 3 appears in a network of chemical reactions. To do this, we need to be able to construct a generative model, as we did above, that accounts for multiple chemical reactions. Figure 4 illustrates an example of a generative model, and associated master equation, that accounts for a system of two (reversible) reactions. The associated reaction constants are given, as functions of the parameters of the generative model, in Table 1. As above, these are obtained by solving for the coefficients of Equation (14) when at steady state. This serves to illustrate two things. First, the methods outlined in the previous section are equally applicable to systems of multiple reactions. Second, when multiple reactions are in play, the generative model can be formulated such that chemical species that do not participate in the same reaction can be treated as being conditionally independent of one another. This induces the sparsity that makes inferential message passing possible. A generic expression of a reaction system obeying the law of mass action is as follows:(22)$$\begin{array}{cc} \partial_{\tau }u_{i}(\tau ) & =\sum_{j}\Omega_{ij}r_{j}(u(\tau ))\\ r_{j}(u(\tau )) & =\kappa_{j}\prod_{i}u_{i}{\left(\tau \right)}^{\sigma_{i}} \end{array}.$$

The stoichiometry matrix **Ω** indicates the difference between *ρ* and *σ* for each chemical species for each reaction. The **r** vector function returns all the reaction rates (treating forwards and backwards reactions as separate reactions). Equation (21) is then a special case of Equation (22) when there are only two reactions. Equation (22) provides a clear depiction of message passing in which each element of **r** is a message, with the stoichiometry matrix determining where those messages are sent. Figure 4 demonstrates the relationship between this message passing and the graphical formulation of chemical reaction systems. Via this graphical notation, Equation (22) has many special cases throughout biology [37,39], some examples of which are outlined in Appendix B. However, for systems with many components, it can be very high-dimensional. The next subsection details one way in which the dimensionality of metabolic networks can be reduced, through an appeal to a separation of time scales.

### 3.3. Enzymes

So far, everything that has been said could apply to any chemical reaction system. However, the introduction of enzymes brings us into the domain of the life sciences. Just as we can group elements on the same side of a reaction to account for autocatalytic dynamics, we can group elements on opposite sides of the reaction to account for catalytic activity. Enzymes are biological catalysts that combine with substrates to form an enzyme–substrate complex, modify the substrate and dissociate from the resulting product. As such, they appear on both sides of the overall reaction.

More formally, an enzymatic reaction has a stoichiometry matrix of the form:(23)$$\begin{array}{l} \Omega =\left[\begin{array}{cccc} \begin{array}{r} -1\\ -1\\ 1\\ 0 \end{array} & \begin{array}{r} 1\\ 1\\ -1\\ 0 \end{array} & \begin{array}{r} 0\\ 1\\ -1\\ 1 \end{array} & \begin{array}{r} 0\\ -1\\ 1\\ -1 \end{array} \end{array}\right]\\ \Rightarrow S_{S}+S_{E}⇌S_{C}⇌S_{P}+S_{E} \end{array}.$$

The rows of **Ω** are the substrate, enzyme, enzyme–substrate complex, and product. These are shown in the reaction system using the subscripts *S*, *E*, *C*, and *P*, respectively. We can express a generative model for this reaction system as follows:(24)$$\begin{array}{cc} P\left(S_{C}⊗S_{E}\right) & =Cat\left(\left[\begin{array}{l} 0\\ \alpha_{1}\\ 1-\alpha_{1}\\ 0 \end{array}\right]\right)\\ P\left(S_{S}|S_{C}⊗S_{E}\right) & =Cat\left(\left[\begin{array}{cccc} \begin{array}{l} 0\\ 1 \end{array} & \begin{array}{l} \alpha_{2}\\ 1-\alpha_{2} \end{array} & \begin{array}{l} 0\\ 1 \end{array} & \begin{array}{l} \alpha_{2}\\ 1-\alpha_{2} \end{array} \end{array}\right]\right)\\ P\left(S_{P}|S_{C}⊗S_{E}\right) & =Cat\left(\left[\begin{array}{cccc} \begin{array}{l} 0\\ 1 \end{array} & \begin{array}{l} 1-\alpha_{2}\\ \alpha_{2} \end{array} & \begin{array}{l} 0\\ 1 \end{array} & \begin{array}{l} 1-\alpha_{2}\\ \alpha_{2} \end{array} \end{array}\right]\right) \end{array}.$$

The first term gives the proportion of enzyme we expect to be in complex form versus being free to engage with the substrate or product. The probability of being in both states simultaneously, and of being in neither of the two states, is zero. When the complex is present, the substrate and product are both absent. When the enzyme is present, the substrate is present with some probability, and the product is present with the complement of that probability. As before, a chemical reaction network can be constructed based upon the conditional independencies of the associated model, i.e., the independence of substrate and product conditioned upon the enzyme and complex, which satisfies the sparse message passing of Equation (23). The requisite rate constants (corrected for the mean-field assumption) are shown in Table 2.

In constructing these messages, we relax the assumption that the steady state is at equilibrium. This means that detailed balance can be violated and involves using a non-zero *β*_1_ from Equation (14) in some of the terms, such that there is constant production of, and removal of, certain chemical species from the system. Specifically, we will assume production of the substrate and removal of the product at equal rate (*c*). A consequence of this is that reactions generating products must be faster than reactions using up product in order for steady state to be maintained. The plots on the left of Figure 5 illustrate the resulting dynamics. Note the initial decrease in substrate and enzyme concentration as they combine to form the complex, followed by the slow rise in product concentration.

In metabolic reaction systems, there are many reactions catalysed by enzymes. In this setting, it is useful to be able to reduce the number of elements of the system to a manageable dimension through omitting the explicit representation of enzymes. A common approach to this is to use Michaelis–Menten kinetics [40]. This depends upon a separation of timescales. The two timescales in question are illustrated in Figure 5 through the rapid conversion of substrate to complex, and the slower generation of product. Combination of substrate and enzyme, and dissociation of complex to substrate and enzyme are both faster than dissociation of complex to product and enzyme. When this is true, a quasi-equilibrium approximation may be adopted. This means that the rates of the reactions involving the substrate are assumed to be much faster than those involving the product:(25)$$\begin{array}{cc} \partial_{\tau }u_{P} & =r(u)-c\\ \begin{array}{c} \kappa_{4}u_{P}u_{E}\ll \kappa_{3}u_{C}\\ \kappa_{1}u_{S}u_{E}\gg \kappa_{3}u_{C}\\ \kappa_{2}u_{C}\gg \kappa_{3}u_{C} \end{array}\} & \Rightarrow r(u)\approx v_{\max}\left(\frac{u_{S}}{\kappa_{m}+u_{S}}\right)\\ v_{\max} & ≜\kappa_{1}\kappa_{3}(u_{E}+u_{C})\\ \kappa_{m} & ≜\frac{\kappa_{2}}{\kappa_{1}}\\ c & \leq (1-z)(1-\alpha_{1}) \end{array}.$$

Equation (25) specifies the quasi-equilibrium assumption [41], and the resulting Michaelis–Menten form for the reaction function **r**. The final line follows from the condition that the rate constants be non-negative. The rate constants, in terms of the generative model, are given in Table 2. This lets us consider what the assumptions underneath the Michaelis–Menten equation mean in relation to the underlying generative model. First, the assumption that the reaction generating the product from the complex is much faster than the reverse reaction implies α_1_ approaches its lower limit. When interpreted from the perspective of the generative model, this makes intuitive sense, as the implication is that given sufficient time, most of the enzyme will be in the non-complex form. Second, the assumption that the forwards and backwards reactions between substrate and complex are faster than the reaction generating the product implies the *z* parameter must be close to one.

Equation (25) simplifies a system comprising four chemical species into a single non-linear rate function that depends only upon the substrate. Practically, this means Michaelis–Menten kinetics can be used to omit explicit representation of enzymes from a reaction system. This lets us replace the reaction function (**r**) from Equation (21) with that from Equation (24), significantly reducing the dimensionality of the resulting system. This formulation is the starting point for methods for further dimensionality reduction of high-dimensional metabolic networks [42]. For instance, the extreme pathway method [43,44] defines extreme pathways as the set of **r** (normalized by *v*_max_ for the rate limiting reaction in the pathway) for which **Ωr** returns a vector of zeros. By taking the singular value decomposition of the matrix of extreme pathway vectors, the left singular vectors can be used to describe ‘eigenpathways.’ By omitting eigenpathways with sufficiently small singular values, a simpler representation of the network may be obtained.

This subsection brought the concept of an enzyme into the chemical kinetics of the previous sections and demonstrated how an appeal to separable timescales results in a simpler, lower-dimensional, representation of an enzymatic system. The associated rate function that resulted from this emphasizes the emergence of nonlinear phenomena at slower scales of a multicomponent system.

### 3.4. Enzymatic Inference

The enzymatic system of the previous subsection is useful in unpacking an active inferential interpretation of chemical kinetics. The plots on the right of Figure 5 are key to this perspective. The upper-right plot illustrates a function of the substrate concentration that converges to the product concentration, and a function of the product concentration that converges to the substrate concentration. There is a sense in which we could interpret this as the substrate, on average, representing beliefs about the product and vice versa [15,25]. The plots of variational free energy (averaged under the enzyme and complex probabilities) decrease over time. The implication is that the models determining the evolution of the substrate and product, both of which predict the enzymatic state, become better explanations for these data (on average) over time.

Although the interactions between the substrate and enzyme are bidirectional, the influence of the enzyme on the product is unidirectional. This is a consequence of the steady state being non-equilibrium. This highlights that there are two ways of optimising free energy. The first is to do as the distributions encoded by the product, and to change beliefs to better explain the data at hand. The second is to do as the substrate does, through changing the data (i.e., enzyme concentration) such that the explanation fits. Note the initial increase in free energy for the model optimized by the product concentration, as the enzyme concentration changes. This is then suppressed, much like a prediction error in neurobiology [45,46,47,48,49], as the product updates its implicit beliefs.

While it might seem a bit strange to formulate the dynamics of one component of a system in relation to a functional of beliefs about other components, this move is central to the Markovian monism that underwrites active inference [50]. It is this that offers us a formal analogy with theoretical neurobiology, and the action-perception loops [51] found in the nervous system. The distinction between active (e.g., muscular) and sensory (e.g., retinal) limbs of these loops derives from the same non-equilibrium property, breaking the symmetry of message passing, such that beliefs can directly influence active but not sensory states. Whether something is active or sensory depends upon the perspective that we take, with enzymes being sensory from the perspective of beliefs encoded by the product, and active from the perspective of beliefs encoded by the substrate.

## 4. Metabolism

In this section, we briefly consider a (fictional) biochemical network that exploits the formulation above. A generative model for the network is illustrated in the upper left of Figure 6. The pink arrows supplement this model with the directional influences assumed at the lowest levels of the model. The lowest level of the model reflects the ‘active’ and ‘sensory’ interactions with another system that is not explicitly modelled here. All reactions are enzymatic, but with explicit treatment of the enzymes omitted via the Michaelis–Menten formulation. As such, the factors corresponding to the enzymes in the model are absorbed into the factors relating the concentrations of reactants. On finding the kinetics consistent with this steady state, the result is the reaction system shown in the upper right of Figure 6.

The plots of the steady state shown in the lower part of Figure 6 use the same layout as the reaction network, but show the marginals (i.e., concentrations) of each species once it reaches steady state. The larger the circle, the greater the concentration. The initial conditions involve zero concentration for all species, so their concentrations can only increase when they receive messages from *S*_3_, via the other reactants.

The lower-left plot shows successful convergence to the non-equilibrium steady state determined by the generative model. The structure of this steady state resembles the architectures found in metabolic networks in the sense that an external system supplies some chemical (*S*_3_) which is converted through a series of reactions into other chemical species (*S*_5_ and *S*_7_) that participate in other reactions external to the system. The glycolysis pathway is one example, in which glucose is provided to the system to be converted to acetyl CoA (taken up by the citric acid cycle) or to lactate (taken up by the Cori cycle) [52].

The lower-right plot in Figure 6 shows the steady state obtained when a lesion is introduced into the message passing, through setting *v*_max_ to zero for the reaction converting *S*_1_ to *S*_4_. Recall that *v*_max_ is a function of the reaction constants which themselves are functions of the parameters of the generative model. For example, when α_1_ approaches its upper limit, the enzyme spends little of its time in complex form, so cannot catalyse the reaction. This effectively induces a disconnection, precluding conversion of *S*_1_ to *S*_4_. The reason for inducing this lesion is to illustrate the diaschisis that results. A diaschisis is a concept from neurobiology [53,54,55]. It refers to the consequences of a localized lesion for distant parts of a network. Just as lesions to one neural system can have wide reaching functional consequences throughout the brain, the consequences of the localized lesion in Figure 6 can be seen throughout the reaction network. In addition to the loss of *S*_4_ and *S*_5_, there is a compensatory increase in *S*_2_ and *S*_6_. This ensures a steady state is attained, as the loss of output from *S*_5_ is offset by increased output from *S*_7_. However, it is not the same steady state as in the pre-lesioned network. A conclusion we can draw from this is that, as in neurobiology [56], a disconnection can be framed as a change to the parameters of a generative model representing the steady state. The distributed message passing that maintains steady state allows for the effects of the disconnection to propagate throughout the network.

One example (of many) of a disorder in which a new steady state is attained following an enzymatic disconnection is due to thiamine deficiency (a.k.a., Beriberi) [57]. Thiamine is a B vitamin that facilitates the action of several important enzymes, including pyruvate dehydrogenase, which converts pyruvate to acetyl CoA. An alternative fate for pyruvate is conversion to lactate [52]. If we were to associate *S*_3_ with glucose, *S*_1_ with pyruvate, *S*_6_ with lactate, and *S*_4_ with acetyl CoA, we could interpret the lesion in Figure 6 as resulting from thiamine deficiency. The resulting accumulation of lactate is consistent with the local increases in this toxic metabolite observed in neural tissue following thiamine depletion [58]. This may be one aspect of the pathophysiology of Wernicke’s encephalopathy and Korsakov’s psychosis [59]. These are forms of ‘dry’ beriberi with profound neurological and psychiatric consequences. While associating this with the lesion in Figure 6 is overly simplistic, it serves to illustrate the way in which the somewhat abstract formulations above could be applied to specific metabolic systems, their disconnections, and the resulting diaschisis.

## 5. Discussion

This paper has sought to apply the probabilistic dynamics that underwrite active inferential approaches to neurobiology to biochemical networks. This started from the expression of a categorical system in terms of a master equation and the interpretation of this equation in terms of flows on a free energy functional. As free energy is a functional of a generative model, this meant the dynamics acquired an interpretation as inference, in the sense of approximating a marginal likelihood. In what followed, the dimensionality of the representation afforded by the master equation was reduced, first through an appeal to a mean-field assumption. The interactions between different factors were simplified by noting that only those variables in the Markov blanket of a given state are necessary to find the appropriate steady-state distribution.

The sparse message passing that resulted from this—reminiscent of the approach used in variational message passing [7]—reduces to the law of mass action under certain assumptions. This lets us treat simple chemical reactions as if they were optimising a generative model. By introducing enzymatic reactions, and working with a non-equilibrium steady state, a further reduction in dimensionality is afforded by Michaelis–Menten kinetics. This emphasizes the emergence of increasingly nonlinear dynamics at higher spatiotemporal scales—something that has been observed in a range of network systems [60,61]. In addition, the combination of the Markov blanket inherent in an enzymatic reaction and the asymmetric message passing in a non-equilibrium system offered an opportunity to frame different parts of the system as optimising beliefs about other parts of the system. This minimisation of free energy through action and ‘perception’ is known as active inference in neuroscience.

Finally, a simple metabolic network was constructed that exploited the reduced expression of enzymatic dynamics, and which utilized the asymmetric message passing associated with active inference. Just as models of inference in the nervous system can be used to simulate pathology through disconnection [62,63,64], this metabolic network was lesioned to illustrate that disconnections, whether axonal or enzymatic, can result in a diaschisis, i.e., distributed changes in distant parts of the network. Crucially, the system still attains steady state following a lesion. It is just a different steady state. This offers a point of connection with approaches in computational neurology [65] and psychiatry [66,67], motivated by the complete class theorems [68,69], which treat pathology as optimally attaining a suboptimal steady state [70]. This perspective places the burden of explanation for pathological behaviour on the prior probabilities associated with the steady state (i.e., it asks ‘what would I have to believe for that behaviour to appear optimal?’). The advantage of this approach is that it provides a common formal language (prior probabilities) in which a range of conditions—from psychosis to visual neglect—can be articulated. The example in Figure 6 suggests metabolic disorders may be amenable to the same treatment.

There are several directions in which the ideas presented in this paper could be pursued. Broadly, these include (i) generalising the dynamics beyond Michaelis–Menten kinetics to include more complex reaction functions, (ii) identifying the generative models of real reaction systems, and (iii) moving beyond metabolic systems to other forms of biological dynamics. Taking these in turn, the Michaelis–Menten formulation can be generalized for molecules (e.g., enzymes) with more than one binding site. This means that there is more than one enzyme–substrate complex state, and a set of reactions allowing transitions between these. One of the most prominent examples is the binding of oxygen to haemoglobin, a protein with four binding sites. The haemoglobin dissociation curve has a sigmoidal form [71], offering an alternative reaction function to the saturating Michaelis–Menten reaction function. More generally, the Hill equation [72] can be obtained using an analogous derivation to the Michaelis–Menten equation and has the latter as a special case.

Identifying generative models in biological chemical networks may be as simple as finding the steady state. However, the perspective offered in Section 3.4 adds an important twist to this. The generative model should express beliefs about something external to the network. To understand the problem a given network is solving, we need to be able to express a model of the inputs to that network. An active inference account of glycolysis would have to start from a generative model of the factors outside of the glycolytic pathway that explain glucose availability. Treating the constituents of the glycolysis pathway as expressing beliefs about the things causing glucose availability, we would hope to find the message passing among elements of the pathway emerge from minimising the free energy of their associated beliefs. Similar approaches have been adopted in neural systems, demonstrating that it is possible to identify implicit probabilistic beliefs about variables in a generative model in networks of in vitro neurons [73]. While outside the scope of this paper, many of these models call for caching of past observations. As highlighted by one of the reviewers, such models need to incorporate forgetting to ensure steady state and preclude convergence to a narrow distribution [74,75].

The above emphasizes what may be the most important practical implication of this paper for metabolic network analysis. Given the scale of such networks in biotic systems, and their interaction with chemical systems in the wider environment, most analyses are restricted to a small part of an open system. In most interesting cases, the kinetics within that system will change when those outside that system change. For instance, the behaviour of a glycolytic network will vary when the rate of lipolysis increases or decreases. This suggests that it should be possible to formulate and test hypotheses of a novel kind. In place of questions about alternative kinetics that could be in play, the inferential perspective lets us ask about the problem a biochemical system is solving, with reference to the probable states of external systems. Practically, this means we can borrow from the (‘meta-Bayesian’ [69]) methods developed in fields such as computational psychiatry—designed to ask questions about the problems the brain is solving—and formalize alternative functional hypotheses about the problem a metabolic network is solving.

There are many biological applications of categorical probabilistic dynamics—sometimes referred to as compartmental models. For instance, in epidemiology [76,77] the movement of people between susceptible, exposed, immune, and recovered compartments mimics the exchanges between different chemical species. Similar mean-field dynamics can be found in neurobiology [78], immunology [79,80], ecology [81], and pharmacokinetics [82]. In addition, they are common outside of biology, in fields such as economics [83] and climate science [84]. In principle, a similar treatment could be applied to such systems, interpreting the interactions between compartments as inferential message passing given a generative model.

## 6. Conclusions

This paper sought to illustrate some points of contact between active inference, a well-established framework in theoretical neurobiology, and the techniques used in modelling biochemical networks. Specifically, the focus was on the relationship between generative models, their associated inferential message passing, and the sparse network interactions in metabolic systems. Under certain assumptions, the master equation describing the evolution of a categorical probability distribution has the same form as the law of mass action, from which standard biochemical results may be derived. This enables construction of a biochemical network, whose rate constants are functions of an underlying generative model. The kinds of pathology affecting this network can be formulated in terms of aberrant prior beliefs, as in computational neurology and psychiatry, and manifest as disconnections whose consequences propagate throughout the network.

## Figures and Tables

**Figure 1 entropy-23-00606-f001:**
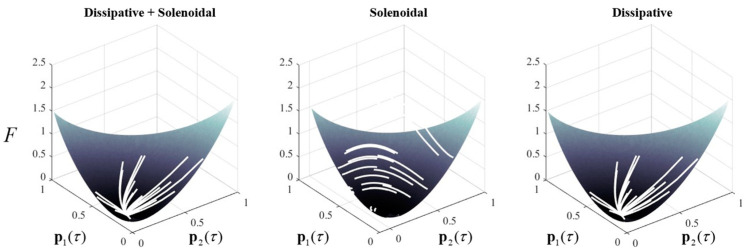
Solenoidal and dissipative dynamics in categorical systems. This figure provides a numerical example of a (three-dimensional) system consistent with Equation (5), and its decomposition as in Equation (6), starting from a series of random initial states. Each trajectory is shown in white. In addition, it illustrates the free energy landscape (in 2 dimensions) to demonstrate the interpretation given in Equation (7). On the left, we see the combination of the dissipative and solenoidal flows that tend towards the free energy minimum. In the centre, the dissipative part of the flow has been suppressed, leading to trajectories around the free energy contours. Such trajectories conserve free energy (but not probability) so do not find its minimum. On the right, the purely dissipative trajectories find the free energy minimum, but take subtly different paths compared to those supplemented with the solenoidal flow.

**Figure 2 entropy-23-00606-f002:**
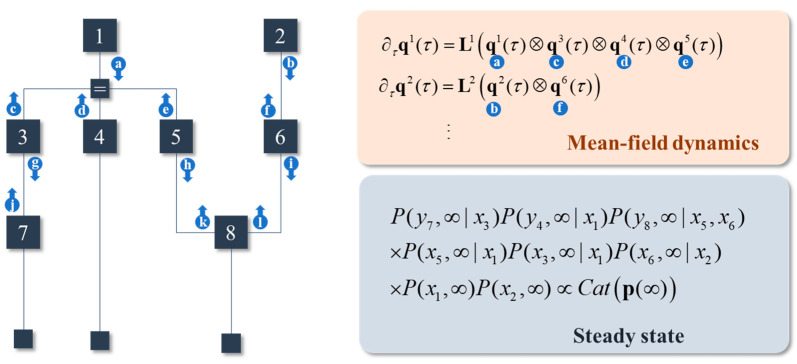
Sparse models and messages. This figure illustrates a generative model using normal (Forney) factor graph. Here, we have 8 different variables. The *y* variables are indicated by the small squares at the bottom of the factor graph. Dependencies between variables, represented on the edges of the graph, are indicated by the square factor nodes. The Markov blanket of a variable is determined by identifying those variables that share a factor (i.e., any edges connected to the associated square nodes). Not every variable is conditionally dependent upon every other; implying this generative model has a degree of sparsity. This lets us simplify the mean-field dynamics such that the rate of change of each marginal distribution depends only upon its Markov blanket. The result has the appearance of message passing, as indicated by the arrows. Each arrow represents a message coming from a factor. Where they meet, they each contribute to the local steady state.

**Figure 3 entropy-23-00606-f003:**
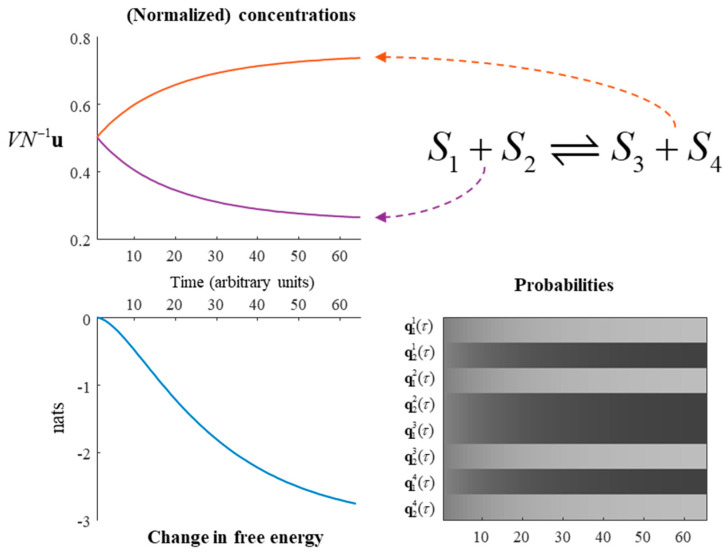
A chemical reaction. This figure illustrates the solution to the generative model outlined in Equation (17), under the dynamics given in Equation (20). The upper-left plot shows the rate of change of the substrates and products. The two substrates have equal concentrations to one another, as do the two products. Under this model, with α = ¼, the substrates are converted into products until the substrates are at a quarter of their maximum concentration, with the remainder converted to the products. The same information is presented in probabilistic form in the lower right. Here, black indicates a probability of 1, white of 0, and intermediate shades represent intermediate probabilities. The plot of free energy over time shows that, despite the mean-field approximation and the constraints applied to the transition rate matrix, the reaction still evolves towards a free energy minimum—as in Figure 1. Note that, in the absence of an external input to this system, the free energy reduces to a Kullback–Leibler divergence between the current state and the steady state.

**Figure 4 entropy-23-00606-f004:**
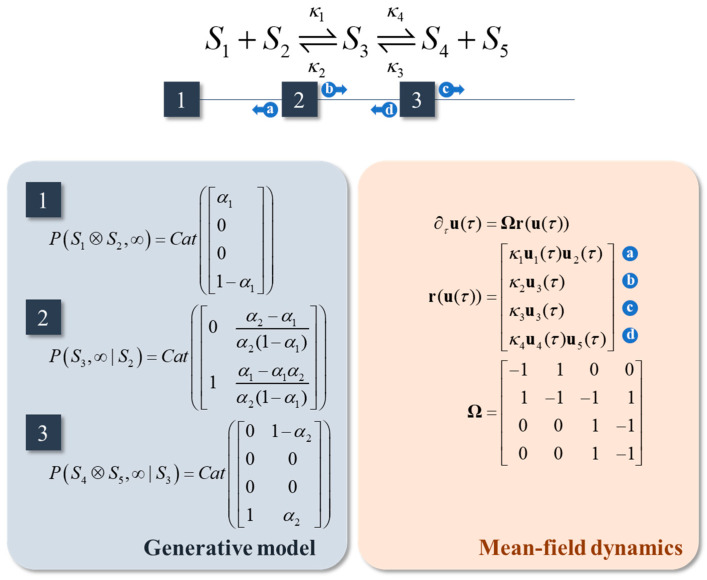
Reaction networks. This schematic illustrates the factor graph associated with a system comprising a pair of coupled reversible reactions (i.e., four reactions in total). The factors are specified in the blue panel. These are chosen to enforce conservation of mass, in the sense that the marginal of *S*_1_ or of *S*_2_ plus the marginal of *S*_3_ plus the marginal of *S*_4_ or of *S*_5_ is one.

**Figure 5 entropy-23-00606-f005:**
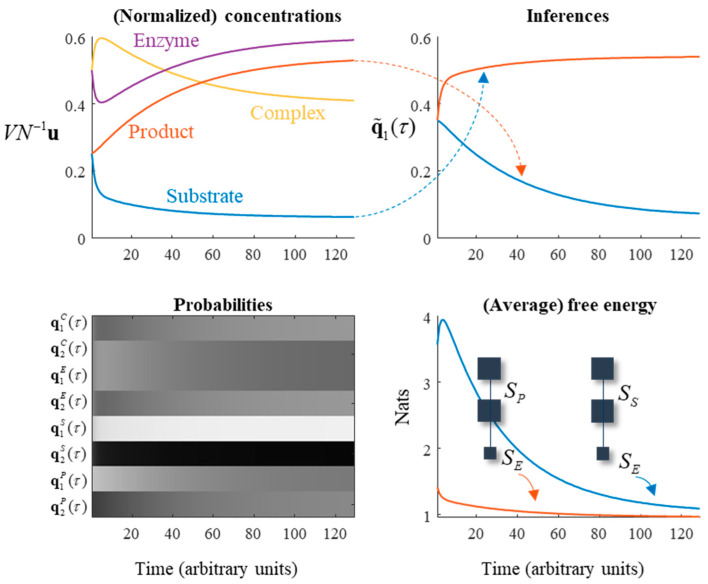
Enzymes, Markov blankets, and chemical inference. This figure illustrates several points. The plots on the left use the same formats as in Figure 3 to show the evolution of the reaction in terms of concentration and probability. The plots on the right exploit the Markov blanket structure implicit in an enzymatic reaction to show the evolution of the ‘beliefs’ implicitly encoded by the expected value of the substrate about the product, and vice versa. The upper-right plot shows these beliefs, defined as ${\tilde{q}}_{1}^{1}=\alpha_{2}-q_{1}^{2}$ and ${\tilde{q}}_{1}^{2}=\alpha_{2}-q_{1}^{1}$, which converge towards $q_{1}^{1}$ and $q_{1}^{2}$, respectively as the steady state is attained. The implicit generative models are shown in the free energy plot, with the enzyme playing the role of the data being predicted. The free energy of each decreases as the beliefs converge upon the posterior probabilities of substrate and product given enzyme.

**Figure 6 entropy-23-00606-f006:**
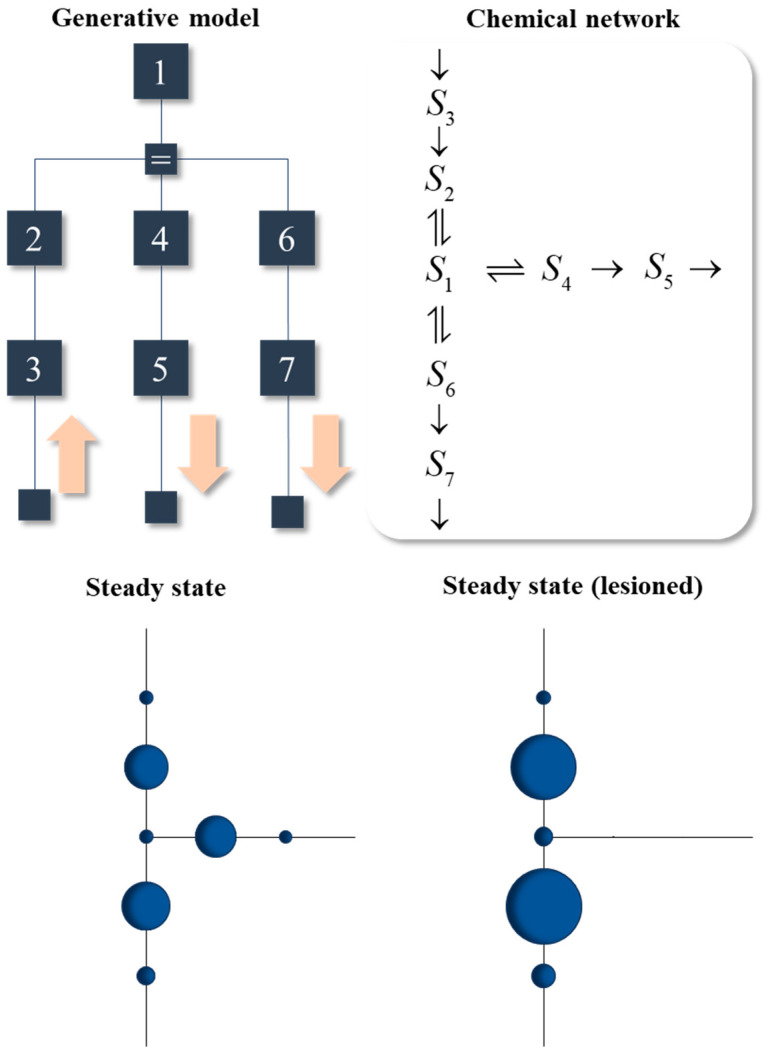
Metabolic networks and their pathologies. This figure shows the conditional dependencies in a generative model in the upper left, highlighting the directional influences at the lowest level of the model with pink arrows. These ensure *S*_3_ is a sensory state, while *S*_5_ and *S*_7_ are active states. In the upper right is the chemical message passing that solves this model. The two plots in the lower part of the figure illustrate the relative probability of the marginal probabilities (or concentrations) of each chemical species. The spatial configuration matches that of the network in the upper right. The sizes of the circles indicate the relative concentrations once steady state has been attained. The plots on the left and right show the steady states before and after introduction of a lesion that disconnects the reaction from *S*_1_ to *S*_4_. Here, we see a redistribution of the probability mass, resulting in an alternative (possibly pathological) steady state.

**Table 1 entropy-23-00606-t001:** Rate constants for the reaction network in Figure 4. This table specifies the *κ* parameters from Figure 4 as functions of the α parameters of the associated generative model and a free parameter *z*.

Rate Constant	Function of α
$\kappa_{1}$	$\lambda N^{-1}Vz\alpha_{1}^{-2}\alpha_{2}^{-1}(\alpha_{2}-\alpha_{1})$
$\kappa_{2}$	λz
$\kappa_{3}$	λ(1−z)
$\kappa_{4}$	$\lambda N^{-1}V(1-z)\alpha_{1}^{-2}\alpha_{2}{\left(1-\alpha_{2}\right)}^{-2}(\alpha_{2}-\alpha_{1})$

**Table 2 entropy-23-00606-t002:** Rate constants for an enzymatic reaction. This table specifies the *κ* parameters from Equation (24) as a function of the α parameters of Equation (23) and a free parameter *z*.

Rate Constant	Function of α
$\kappa_{1}$	$\lambda N^{-1}V(z-\alpha_{1}z+c)\alpha_{1}^{-2}\alpha_{2}^{-1}$
$\kappa_{2}$	λz
$\kappa_{3}$	λ(1−z)
$\kappa_{4}$	$\lambda N^{-1}V((1-z)(1-\alpha_{1})-c)\alpha_{1}^{-2}{\left(1-\alpha_{2}\right)}^{-1}$

## Data Availability

The Matlab code used to generate the figures in this paper is available at https://github.com/tejparr/Message-Passing-Metabolism accessed on 12 May 2021.

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
