# Peer review of "Message Passing and Metabolism"

_entropy, 2021, doi:10.3390/e23050606_

Round 1
Reviewer 1 Report
This paper presents an interesting set of homologies between the treatment of metabolism and neurobiology in terms of their treatment with stochastic dynamics. The results are properly and accurately described and the writing is concise and to the point. As such I recommend the paper for publication in Entropy.
As more of a specialist in statistics in neurobiology, however, I have two suggestions which might make it easier for other readers to appreciate the import of these homologies.
The first would be to provide an overview of the flow of the argument in less technical terms at the start of the paper, perhaps after the introduction. While the overall flow is explained in more technical concise language, I suspect many readers would benefit from a less technical explanation of how this paper will proceed.
The second is whether there is some application of these homologies which leads to new specific knowledge in the field of metabolism? The paper provides some brief abstract examples, but it strikes this reviewer that a specific application generating a new finding, even if fairly simple, would add to the impact of the paper.
Author Response
This paper presents an interesting set of homologies between the treatment of metabolism and neurobiology in terms of their treatment with stochastic dynamics. The results are properly and accurately described and the writing is concise and to the point. As such I recommend the paper for publication in Entropy.
Many thanks for this evaluation.
As more of a specialist in statistics in neurobiology, however, I have two suggestions which might make it easier for other readers to appreciate the import of these homologies.
The first would be to provide an overview of the flow of the argument in less technical terms at the start of the paper, perhaps after the introduction. While the overall flow is explained in more technical concise language, I suspect many readers would benefit from a less technical explanation of how this paper will proceed.
Thank you, I have added the following to the introduction:
The argument of this paper can be overviewed as follows. Given an interpretation of a steady state as a generative model, the behavior of stochastic systems that tend towards that distribution can be interpreted as inference. When such systems are formulated in terms of master equations, they have the appearance of gradient flows on a free energy functional—the same functional used to score approximate Bayesian inference schemes. In practice, we may be interested in high-dimensional systems, for which the distribution can be factorized into a set of lower-dimensional systems. In this setting, a mean-field approximation can be applied such that we only need work with these lower-dimensional marginal distributions. When the implicit generative model is sufficiently sparse, steady state can be achieved through dynamics that involve sparse coupling between the marginal distributions—analogous to inferential message passing schemes in machine learning. Under certain assumptions, these probabilistic dynamics have the appearance of chemical kinetics, licensing an interpretation of some chemical reactions—including biological, enzymatic, reactions—as if the chemical species were engaged in a form of (active) inference about one another. An implication of this interpretation is that metabolic pathologies can be framed in terms of the implicit generative model (i.e., steady state) they appear to be solving.
The second is whether there is some application of these homologies which leads to new specific knowledge in the field of metabolism? The paper provides some brief abstract examples, but it strikes this reviewer that a specific application generating a new finding, even if fairly simple, would add to the impact of the paper.
Thank you for this comment. As a non-expert in metabolism, I am cautious about claiming to have discovered anything new in this field, but I have added the following to highlight practical implications for metabolic network analysis:
The above emphasizes what may be the most important practical implication of this paper for metabolic network analysis. Given the scale of such networks in biotic systems, and their interaction with chemical systems in the wider environment, most analyses are restricted to a small part of an open system. In most interesting cases, the kinetics within that system will change when those outside that system change. For instance, the behavior of a glycolytic network will vary when the rate of lipolysis increases or decreases. This suggests that it should be possible to formulate and test hypotheses of a novel kind. In place of questions about alternative kinetics that could be in play, the inferential perspective lets us ask about the problem a biochemical system is solving, with reference to the probable states of external systems, in terms of generative models. Practically, this means we can borrow from the (‘meta-Bayesian’ [70]) methods developed in fields like computational psychiatry—designed to ask questions about the problems the brain is solving—and formalize alternative functional hypotheses about the problem a metabolic network is solving.
Reviewer 2 Report
I thoroughly enjoyed the paper "Message passing and metabolism". The author shows convincingly that an active inference approach – a theoretical approach having at its core a Bayesian perspective – can be extended to fields other than neurobiology, where it is quite influential. The appeal of this approach is that it allows for original interpretations of metabolic networks and potentially many other biological systems in terms of message passing and flow of information. While many of the core concepts were explored before, I think the paper certainly deserves publication for several reasons.
From the point of view of the content, it nicely integrates the concept of Markov blankets and networks of categorical states, providing a glimpse of how this approach could be leveraged to examine real world metabolic networks. Many systems include sparse networks, so I think that the treatment presented here is a nice illustration of some basic principles that could be applied with efficacy in different contexts. Besides, it is very well written, and the author take his time in explaining clearly basic concepts of active inference. I only found some passages a bit unexplained and where terms were used a bit out of the blue, like the section relative to Figure 1, in which there are also some very minor mistakes and typos ( e.g. dissipitive rather than dissipative in Figure 1 ). I believe these small mistakes could be corrected in the proofs.
The major question left after reading the paper is "yes, I see why the perspective is interesting. But how does this approach translate in an actual benefit or tool to analyze metabolic networks?". I wish more was said about this. I do not think it is fundamental though.
Author Response
I thoroughly enjoyed the paper "Message passing and metabolism". The author shows convincingly that an active inference approach – a theoretical approach having at its core a Bayesian perspective – can be extended to fields other than neurobiology, where it is quite influential. The appeal of this approach is that it allows for original interpretations of metabolic networks and potentially many other biological systems in terms of message passing and flow of information. While many of the core concepts were explored before, I think the paper certainly deserves publication for several reasons.
Many thanks for this.
From the point of view of the content, it nicely integrates the concept of Markov blankets and networks of categorical states, providing a glimpse of how this approach could be leveraged to examine real world metabolic networks. Many systems include sparse networks, so I think that the treatment presented here is a nice illustration of some basic principles that could be applied with efficacy in different contexts. Besides, it is very well written, and the author take his time in explaining clearly basic concepts of active inference. I only found some passages a bit unexplained and where terms were used a bit out of the blue, like the section relative to Figure 1, in which there are also some very minor mistakes and typos ( e.g. dissipitive rather than dissipative in Figure 1 ). I believe these small mistakes could be corrected in the proofs.
Thank you for spotting these typos—I have ensured they are corrected.
The major question left after reading the paper is "yes, I see why the perspective is interesting. But how does this approach translate in an actual benefit or tool to analyze metabolic networks?". I wish more was said about this. I do not think it is fundamental though.
Thank you for this. Although the primary focus of this paper is construct validation, I fully agree with the importance of practical application. As such, I have added the following the discussion:
The above emphasizes what may be the most important practical implication of this paper for metabolic network analysis. Given the scale of such networks in biotic systems, and their interaction with chemical systems in the wider environment, most analyses are restricted to a small part of an open system. In most interesting cases, the kinetics within that system will change when those outside that system change. For instance, the behavior of a glycolytic network will vary when the rate of lipolysis increases or decreases. This suggests that it should be possible to formulate and test hypotheses of a novel kind. In place of questions about alternative kinetics that could be in play, the inferential perspective lets us ask about the problem a biochemical system is solving, with reference to the probable states of external systems. Practically, this means we can borrow from the (‘meta-Bayesian’ [70]) methods developed in fields like computational psychiatry—designed to ask questions about the problems the brain is solving—and formalize alternative functional hypotheses about the problem a metabolic network is solving.
Reviewer 3 Report
Message Passing And Metabolism by Thomas Parr is a novel paper translating the concepts of active inference from theoretical neurobiology to chemical reaction kinetics.
In the paper the author introduces the basic concept of the notion of message passing and then applies it to some fundamental biochemical reaction networks to explain its use.
The concept of generative models in reaction kinetics community is novel, hence, why this reviewer thinks this paper should be accepted.
The reviewer however has two simple requests for the author, aimed at helping the reaction kinetics community more easily see the connection.
Firstly, can the author please elaborate Equation 19, Table 1, and Table 2, to very simply explain how these corrections were derived.
Secondly, the model in Figure 6 already has a generative model with respect to its stationary distribution, which was derived through the chemical master equation (doi: 10.1007/s00285-006-0034-x). It would be nice to see a comparison of the two in the appendix, or perhaps a short discussion on comparing the two different approaches.
On a stylistic note, the reviewer would like to suggest to the author not to switch between application topics through the paper. It is understandable that the intuition comes from neurobiology, hence, the urge to keep going back to this field to explain the intuition. However, given the intended reader base are from chemical kinetics, it would be good to keep the intuition in the context of chemical kinetics. This is only a comment, for the authors future work. Other than that, it was a really well written paper.
Author Response
Message Passing And Metabolism by Thomas Parr is a novel paper translating the concepts of active inference from theoretical neurobiology to chemical reaction kinetics.
In the paper the author introduces the basic concept of the notion of message passing and then applies it to some fundamental biochemical reaction networks to explain its use.
The concept of generative models in reaction kinetics community is novel, hence, why this reviewer thinks this paper should be accepted.
Many thanks for this.
The reviewer however has two simple requests for the author, aimed at helping the reaction kinetics community more easily see the connection.
Firstly, can the author please elaborate Equation 19, Table 1, and Table 2, to very simply explain how these corrections were derived.
Apologies for this omission. I have added:
We can correct for the discrepancy by raising the numerator of β to a power of the number of substrates, and the denominator to the power of the number of reactants. This correction is obtained by setting Equation 14 to zero when the marginals are consistent with those at steady state and solving for the β coefficients.
And:
The associated reaction constants are given, as functions of the parameters of the generative model, in Table 1. As above, these are obtained by solving for the coefficients of Equation 14 when at steady state.
Secondly, the model in Figure 6 already has a generative model with respect to its stationary distribution, which was derived through the chemical master equation (doi: 10.1007/s00285-006-0034-x). It would be nice to see a comparison of the two in the appendix, or perhaps a short discussion on comparing the two different approaches.
Many thanks for drawing my attention to this paper. I have added the following to the appendix to discuss this:
An excellent example of the application of the chemical master equation, highly relevant to the treatment in this paper, is given in [86]. Focusing on monomolecular reaction systems, the authors detail the relationship between steady state (i.e., the implicit generative model) and the associated reaction kinetics. Their results highlight the way in which a steady state can be determined from the kinetics. This complements the approach pursued here, in which the kinetics, under certain assumptions, emerge from the steady state.
On a stylistic note, the reviewer would like to suggest to the author not to switch between application topics through the paper. It is understandable that the intuition comes from neurobiology, hence, the urge to keep going back to this field to explain the intuition. However, given the intended reader base are from chemical kinetics, it would be good to keep the intuition in the context of chemical kinetics. This is only a comment, for the authors future work. Other than that, it was a really well written paper.
Thank you—I will bear this in mind for the future.
Reviewer 4 Report
This paper evokes biological advantage of the active inference, and applies the extension of the active inference to biochemical networks.
Although it could play an essential role in theoretical biology, the author should refer to the studies on inverse Bayesian inference.
Both active inference and inverse Bayesian inference are motivated by pointing out the disadvantage of Bayesian inference. Since Bayesian inference can contract the probability space by replacing P(h) with P(h|d), it could be converged to local optimum. In engineering problem, the probability space, the potential, or the context are fixed and invariant. Thus, accessibility to the global optimum is very important, and some techniques like simulated annealing were proposed.
In contrast, in biological or psychological problem, the context and/or the potential is not fixed, and they are perpetually changed. Therefore, accessibility to the global optimum does not make sense.
However, dissipation from the local optimum is required in biological or psychological system, because the local optimum itself is instable. Inverse Bayesian inference improves this problem by the likelihood of the hypotheses are perpetually changed dependent on the systems’ own experience. While this idea is very similar with the idea of active inference, there is priority in inverse Bayesian inference. The author should refer to these studies in Introduction. I recommend the minor revision.
Author Response
This paper evokes biological advantage of the active inference, and applies the extension of the active inference to biochemical networks.
Although it could play an essential role in theoretical biology, the author should refer to the studies on inverse Bayesian inference.
Both active inference and inverse Bayesian inference are motivated by pointing out the disadvantage of Bayesian inference. Since Bayesian inference can contract the probability space by replacing P(h) with P(h|d), it could be converged to local optimum. In engineering problem, the probability space, the potential, or the context are fixed and invariant. Thus, accessibility to the global optimum is very important, and some techniques like simulated annealing were proposed.
In contrast, in biological or psychological problem, the context and/or the potential is not fixed, and they are perpetually changed. Therefore, accessibility to the global optimum does not make sense.
However, dissipation from the local optimum is required in biological or psychological system, because the local optimum itself is instable. Inverse Bayesian inference improves this problem by the likelihood of the hypotheses are perpetually changed dependent on the systems’ own experience. While this idea is very similar with the idea of active inference, there is priority in inverse Bayesian inference. The author should refer to these studies in Introduction. I recommend the minor revision.
Thank you for this suggestion. I have added:
While outside the scope of this paper, many of these models call for caching of past observations. As highlighted by one of the reviewers, such models need to incorporate forgetting to ensure steady state and preclude convergence to a narrow distribution [75, 76].